# Efficacy of HE4, CA125, Risk of Malignancy Index and Risk of Ovarian Malignancy Index to Detect Ovarian Cancer in Women with Presumed Benign Ovarian Tumours: A Prospective, Multicentre Trial

**DOI:** 10.3390/jcm8111784

**Published:** 2019-10-25

**Authors:** Vincent Dochez, Mélanie Randet, Céline Renaudeau, Jérôme Dimet, Aurélie Le Thuaut, Norbert Winer, Thibault Thubert, Edouard Vaucel, Hélène Caillon, Guillaume Ducarme

**Affiliations:** 1Service de Gynécologie-Obstétrique, CHU de Nantes, 44093 Nantes, France; norbert.winer@chu-nantes.fr (N.W.); Thibault.thubert@chu-nantes.fr (T.T.); edouard.vaucel@chu-nantes.fr (E.V.); 2Service de Gynécologie-Obstétrique, Centre Hospitalier de Saint-Nazaire, 44606 Saint-Nazaire, France; m.randet@ch-saintnazaire.fr; 3Service de Gynécologie-Obstétrique, Centre Hospitalier, 49325 Cholet, France; cline1484@hotmail.com; 4Centre de Recherche Clinique, Centre Hospitalier Départemental Vendée, 85925 La Roche sur Yon, France; jerome.dimet@ght40.fr (J.D.); Aurelie.LETHUAUT@chu-nantes.fr (A.L.T.); 5Service de Biochimie, CHU de Nantes, 44093 Nantes, France; helene.caillon@chu-nantes.fr; 6Service de Gynécologie-Obstétrique, Centre Hospitalier Départemental Vendée, 85925 La Roche sur Yon, France; guillaume.ducarme@chd-vendee.fr

**Keywords:** HE4, CA125, ovarian cancer, presumed benign ovarian tumour, RMI, ROMA

## Abstract

Background: Presumed benign ovarian tumours (PBOT) are defined by the International Ovarian Tumour Analysis (IOTA) group, without suspected sonographic criteria of cancer, without ascites or metastasis. The aim is to evaluate the efficacy of human epididymis protein 4 (HE4), cancer antigen 125 (CA125), the risk of malignancy index (RMI) and the risk of ovarian malignancy index (ROMA) to predict ovarian cancer in women with PBOT. Methods: It is a prospective, observational, multicentre, laboratory-based study including women with PBOT in four hospitals from 11 May 2015 through 12 May 2016. Preoperative CA125 and HE4 plasma levels were measured for all women. The primary endpoint was the specificity of CA125 and HE4 for diagnosing ovarian cancer. The main secondary endpoints were specificity and likelihood ratio of RMI, ROMA and tumours markers. Results: Two hundred and fifty patients were initially enrolled and 221 patients were finally analysed, including 209 benign ovarian tumours (94.6%) and 12 malignant ovarian tumours (5.4%). The malignant group had significantly higher mean values of HE4, CA125, RMI and ROMA compared to the benign group (*p* < 0.001). Specificity was significantly higher using a combination of HE4 and CA125 (99.5%) compared to either HE4 or CA125 alone (90.4% and 91.4%, respectively, *p* < 0.001). Moreover, the positive likelihood ratio for combination HE4 and CA125 was significantly higher (104.5; 95% CI 13.6–800.0) compared to HE4 alone (5.81; 95% CI 2.83–11.90) or CA125 alone (6.97; 95% CI 3.91–12.41). Conclusions: The combination of HE4 and CA125 represents the best tool to predict the risk of ovarian cancer in patients with a PBOT.

## 1. Background

Ovarian cancer is the 5th worldwide leading cause of death for women with cancer [1], often diagnosed in the majority of cases at an advanced stage. The mean survival rate is only 50% at five years [2]. For ovarian lesions, more than 90% are detected before menopause, and over 60% of those detected after menopause are subsequently found to be benign [3]. Early identification of either malignant or benign ovarian tumours is essential [4]. Recently, guidelines by the French National College of Obstetricians and Gynaecologists (CNGOF) concerning ovarian tumours introduced the term of presumed benign ovarian tumour (PBOT) instead of ovarian lesion or ovarian cyst [5,6]. PBOT is an ovarian tumour (more often a cyst but not always) that is not classified by ultrasound findings as a polycystic ovary nor a malignant or borderline tumour, and whose diagnosis requires complementary explorations (imaging examination, serum markers like Carbohydrate Antigen 125) that must be prioritized in order to propose an adapted management (abstention, monitoring, surgical treatment) [5,6,7].

Several serum markers have been evaluated. The Carbohydrate Antigen 125 (CA125) measurement is not very sensitive in the early stage of ovarian cancer [8], and high levels of CA125 are also reported in other physiological or pathological conditions such as pregnancy, menstruation or endometriosis [9]. Then, the efficacy of CA125 serum level is insufficient for diagnosing malignant tumour in PBOT. Other biomarkers were developed to improve specificity for ovarian carcinomas diagnosis such as Human Epididymis Protein 4 (HE4) [8], which is only slightly expressed in the epithelium tissues of respiratory and reproductive organs but overexpressed in ovarian tumours (especially in serous and endometrioid ovarian carcinomas) [10]. Yanaranop et al. [11] reported a specificity of 86% for HE4 and the receiver operating characteristic (ROC) area under the curve (AUC) demonstrating a better efficacy for HE4 alone compared to CA125 alone to diagnose ovarian cancer (0.893 and 0.865, respectively) [12]. It is possible to interpret either the values of these markers independently or by associating them within mathematical formulas. Two algorithms are often used as well: The risk of malignancy index (RMI) [13] and the risk of ovarian malignancy algorithm (ROMA) [14]. They combine both values of CA125 and HE4 with menopausal status. While these markers are quite specific, they are not really sensitive. Guidelines were based on studies with ovarian tumours with or without evidence of underlying ovarian cancer [15,16]. Indeed, these studies included all patients presenting for an ovarian tumour including suspected malignant tumours with or without evidence of metastasis. There is therefore a recruitment bias in relation to this specific population. To our knowledge, no other published study has evaluated serum markers in women with PBOT and without advanced obvious ovarian cancers. Our aim was to evaluate efficacy of CA125, HE4, RMI and ROMA algorithms to predict ovarian cancer in women presenting PBOT.

## 2. Materials and Methods

We conducted a prospective, observational, multicentre, laboratory-based study concerning biomarkers and ovarian tumours, including four centres in the west of France: one university hospital (Nantes) and 3 general hospitals (Saint Nazaire, Cholet and La Roche sur Yon). Women with a PBOT (without suspected sonographic criteria of polycystic ovary or cancer, without ascites or metastasis) and planned surgery were included. The exclusion criteria were pregnant women, age <18 years-old, patient under guardianship, trusteeship or deprived of liberty, renal failure, women with ascites or metastases or malignant ovarian mass presumed under the rules of International Ovarian Tumour Analysis (IOTA) group (if one or more M features are present in absence of B feature, mass is classified as malignant) [4], and women with a suspected adnexal torsion. Indeed, we considered it inappropriate to include women with suspicion of torsion, since regardless of the result of the tumour markers, the patient had to be operated to remove the torsion of the ovary and preserve the ovarian tissue.

All participants were informed about the study and received written information by the surgeon when planned for laparoscopic surgery. Written consent was obtained by the surgeon before inclusion. This study protocol and this consent procedure were approved by the Research Ethics Committee Ouest Nantes IV, France (n° 2014-A01522-45, obtained April 7, 2015) before the beginning of the study. All methods were performed in accordance with the relevant guidelines and regulation. All participants underwent the informed consent process using institutional review board-approved consent documents. This trial is registered with clinicaltrials.gov (NCT02326064).

During this pre-inclusion visit, demographic characteristics, examination and ultrasound criteria were verified to evaluate the different inclusion and exclusion criteria. Menopausal status was defined as the absence of menstruation for more than 6 months, the presence of clinical signs of menopause, or if they were over 50 years of age with a history of hysterectomy. On the day of surgery, an additional blood sample was collected during the preoperative evaluation to perform CA125 and HE4 assays. Blood was collected in a standard heparinized vial. Samples were sent to a central laboratory unit (biochemistry laboratory at the Nantes university hospital), centrifuged and plasmas were stored at −20 °C until analysis. Plasma CA125 and HE4 concentrations were determined by run in single measurements using an electrochemiluminescence Elecsys immunoassay (ECLIA) on a Roche Diagnostics Cobas 8000^®^ e602 analyser (Roche Diagnostics, Mannheim, Germany). The ECLIA assay is a highly sensitive and selective immunologic method. For CA125 and HE4, two types of antibodies were respectively used to make a “sandwich”: One antibody recognizing the analyte was labeled with ruthenium and the other with biotin. With the addition of streptavidin, then of tripropylamine (TPA) and the application of electrode potential, ruthenium was excited, leading to photon emission. The luminescence signal measured was proportional to the analyte concentration, which was obtained using a calibration curve. Prior to measuring, each run was validated by testing two quality control levels. These assays demonstrated high level of correlations with total imprecision CV below 5%. The manufacturer evaluates the quantification limit at 2 UI/mL for CA125 and 5 pmol/L for HE4. Samples with levels above the limit were appropriately diluted according to the manufacturer instructions. The CA125 reference values were less than 35 U/mL [17]. Regarding HE4 values, we used predetermined thresholds for menopausal status: less than or equal to 70 pmol/L for non-menopausal women and less than or equal to 140 pmol/L for postmenopausal women [18,19,20].

Risk of malignancy index (RMI) was calculated according to the formula: RMI = U × M × CA125 with U = ultrasound score (U = 0 if ultrasound score = 0, U = 1 if ultrasound score = 1, U = 3 if ultrasound score 2 to5), M = menopause status (M = 1 for pre-menopausal women, M = 3 for post-menopausal women) [13]. An RMI score greater than 200 was strongly correlated with a high risk of malignancy (sensitivity = 85.4% and specificity = 96.9%) [21].

The risk of ovarian malignancy algorithm (ROMA) score corresponds to predicted probability and is expressed by a percentage rate [14]. Based on the immunoassay for CA125 and HE4, the thresholds may differ to categorize patients in a low or high risk group [22]. In fact, with the Roche Diagnostics Laboratory’s ECLIA method, the cut off level for classifying patients at a high risk is 11.4% for pre-menopausal patients, and 29.9% for menopausal patients.

A prospective database of women included in the study was established. Maternal sociodemographic and clinical characteristics were collected prospectively by a research assistant, independent of the local medical team. The final histological findings were recovered. Serum markers results were blinded to the surgeon and the histopathological analysis were received 1 month after surgery, by a gynaecologic pathologist. Borderline tumours are characterized by the absence of invasive implants on pathological examination, as for benign tumours, but by a proliferation which can be important, found in malignant tumours. Borderline tumours can be considered as pre-cancerous conditions. It therefore seems necessary to classify borderline tumours as malignant tumours and not benign tumours. Moreover, in most studies evaluating the interest of tumour markers CA125 or HE4, the authors classify borderline tumours in the malignant group. We therefore compared benign tumours on one side versus malignant tumours on the other side, including borderline tumours.

The primary endpoint was the specificity of CA125 and HE4 for the diagnosis of ovarian cancer in women with PBOT. Secondary endpoints were the specificity of RMI and ROMA, and AUC of ROC of these markers and algorithms.

### 2.1. Sample Size 

The number of subjects needed for this study was based on the literature. In a meta-analysis [16], HE4 and CA125 had the same sensitivity of 79% (95% CI 0.76–0.81, and 0.77–0.82, respectively), and a higher specificity was found for HE4 93%, (95% CI 0.92–0.94) compared to CA125 (78%, 95% CI 0.76–0.80). This difference was also observed in another meta-analysis [23], with a specificity for HE4 of 94% (95% CI 0.90–0.96) and 78% for CA125 (95% CI 0.73–0.83). We aimed to demonstrate a difference in specificity between HE4 and CA125 of at least 10% (94% and 84%, respectively). With a power of 80% (β error = 0.20), an alpha risk of 5% (α error = 0.05) and knowing that the expected prevalence of ovarian cancers is 8.3%, the number of patients needed was estimated at 162. The recruitment capacities of the four centres made it possible to consider the inclusion of 250 patients over 18 months.

### 2.2. Statistical Analysis

Baseline characteristics were described as number (percentage) for categorical variables and compared between benign and borderline/malign tumour using chi-square tests (or Fisher exact tests as appropriate). Quantitative were described as mean (standard deviation) or median (interquartile range) and compared using *t*-tests (or Mann–Whitney tests as appropriate).

Specificities of CA125 and HE4 (primary endpoint) were estimated with 95% confidence interval and both compared with a Mac-Nemar paired Chi-2 test.

Diagnostic accuracy of serum markers (CA125, HE4) and algorithms (RMI, ROMA) were estimated using the area under the receiver operating characteristic curve (AUC), with the 95% confidence interval (secondary endpoints). Diagnostic accuracy of these markers was compared using Delong test for correlated receiver operating characteristic curves [24].

Sensitivity, negative and positive predictive value and likelihood ratio were estimated with 95% confidence interval.

No imputation was performed missing data. Patients without gold standard (histological data), will not be included in the analysis.

All tests were 2-tailed. *p* values of less than 0.05 were considered significant. Analyses were performed using Stata Software Release 13 (StataCorp LP, College Station, TX, USA).

## 3. Results

Two hundred and fifty women were initially included in our study in four centres from May 11, 2015 to May 12, 2016 (Figure 1). Six women were incorrectly included due to non-compliance with inclusion criteria: two for ultrasound ascites, two for presumed ovarian malignant mass using IOTA rules, one woman was minor and one woman had plasma BetaHCG still positive within one month after abortion. In addition, 23 women were excluded: six who did not undergo surgery (patient request), 10 who underwent surgery but no cyst was observed during laparoscopy, five without histological examination and two without serum markers analysis before surgery. The study population was therefore 221 women.

Among the 221 patients, there were 209 (94.6%) benign and 12 (5.4%) malignant ovarian tumours (two adenocarcinomas and 10 borderline tumours) (See Appendix A). Serous cystadenoma and mature teratoma were the most frequent benign histological type. Among the 10 borderline tumours, we found seven serous tumours, two mucinous tumours and one endometrioid tumour. Age, body mass index (BMI) or menopausal status were not significantly different between benign and malignant tumours (Table 1).

Mean values of HE4, CA125 RMI and ROMA were significantly higher in malignant group than in the benign group (*p* < 0.001) (Figure 2 and Table 1). For the RMI evaluation, three women were excluded because the ultrasound report was not sufficiently documented to calculate it. This score was then established with 218 women. Among these three women, there were two women in the malignant/borderline group (two women with borderline tumours) and one woman in the benign group.

Specificity for CA125, HE4, RMI and ROMA was 90.4%, 91.4%, 99.0% and 83.3%, respectively (See Appendix A). To determine the specificity of the association of HE4 and CA125, we considered as abnormal a CA125 score above the threshold of 35 U/L associated with an HE4 value greater than 70 pmol/L in premenopausal patients and 140 pmol/L in postmenopausal patients. If at least one of the two markers were below the thresholds, the result was considered normal. Specificity was significantly higher using a combination of HE4 and CA125 (99.5%) compared to HE4 or CA125 alone (90.4% and 91.4%, respectively, *p* < 0.001). Specificity of RMI algorithm was significantly higher than ROMA (99.0% and 83.3%, respectively, *p* < 0.001), but was not significantly different compared to a combination of HE4 and CA125 (*p* = 0.99) (Table 2).

The area under the receiver operating characteristic curve for CA125, HE4, association of CA125 and HE4, RMI and ROMA were not significantly different (0.83, 0.91, 0.92, 0.88 and 0.92, respectively) (See Appendix A). Moreover, the positive likelihood ratio for combination HE4 and CA125 was significantly higher (104.5; 95% CI 13.6–800.0) compared to HE4 alone (5.81; 95% CI 2.83–11.90), CA125 alone (6.97; 95% CI 3.91–12.41) or ROMA (4.48; 95% CI 2.87–6.99).

## 4. Discussion

In our study, we demonstrated that the analysis of plasma markers CA125 and HE4, either alone or as part of an algorithm, was useful to confirm diagnosis in the context of PBOT as defined by IOTA rules when these are elevated. Patients are selected after ultrasound echography with presumed benign ovarian tumour. This technique has demonstrated a very good sensitivity and can detect a large number of women at risk of cancer without missing out [4]. So, in this study, in patients with suspected benign cancer, we would like to study whether these markers can be used to identify women with benign cancers and avoid misrepresenting them as having malignant cancer (markers with good specificity).

In our study, the specificity of CA125 for the detection of ovarian cancers is higher than that found in the literature (90.4% (95% CI 85.6–94.1) vs. 78% (95% CI 76–80)) [16]. This difference may probably be explained by our inclusion characteristics. Several studies find a prevalence of ovarian cancer up to 30% or even 53% [25,26]. These proportions are more important than in our population (5.4%) because we excluded all ovarian tumours associated with ascites or metastases or tumours with ultrasound criteria of malignancy using IOTA rules. Indeed, ultrasound has played the role of sensitivity. One of the main purposes of a screening test is to avoid the number of false negatives, i.e., the test is negative while the patient has a malignant cyst. It must therefore have a high sensitivity. This sensitivity is played by the ultrasound, because it allows to identify all the patients having an ovarian cyst and thus potentially ovarian cancer. In fact, patients with ovarian cancer without cysts are very rare. We must find another test to identify in this target population—patients at higher risk of developing ovarian cancer. This is why we have studied the specificity of the tests, in order to avoid the number of false positives, that is to say a positive test while the patient has a benign lesion. Then, it is difficult to compare our results concerning the specificity with other findings in the literature. To our knowledge, there is no study which has evaluated serum markers in women with PBOT.

Regarding the cut-off value of HE4 used and menopausal status, clinicians have the choice [27,28]. Indeed, these values can be used regardless of the immunological method performed. Nevertheless, since there is evidence that HE4 increases significantly during age [29,30], the use of 2 cut-off (70 and 140 pmol/L) seems preferable to the use of a single cut-off (140 pmol/L). The use of these two thresholds is also suggested in a recent study [31]. Nevertheless, compared to the study by Yanararop et al. using the same cut-off as in our study, a greater specificity was found in our study (91.4% vs. 86%) [11].

Several studies have evaluated the combination of CA125 and HE4 concentrations. Using the ECLIA method with a threshold value for HE4 of 140 pmol/L, Chen et al. reported a specificity of 65.7% [32]. In a different study using another technique to measure HE4, the specificity of CA125 and HE4 together was much better (80%) than each alone [33]. In our study, a specificity of 99.5% (97.4–100%) was found for the association of CA125 and HE4 and was significantly higher than those for CA125 or HE4 alone (*p* < 0.001). Moreover, the AUC for CA125 and HE4 association is higher in our study (0.92, 95% CI 0.85–0.99) than AUC for CA125 or HE4 alone (0.83, 95% CI 0.68–0.97 and 0.91, 95% CI 0.85–0.96, respectively). This AUC is comparable to the data in the literature: 0.96 (95% CI 0.93–1.00) in the study by Chen et al. [32] and 0.91 (95% CI 0.86–0.96) in the study by Moore et al [34]. The combination of the CA125 and HE4 assays is therefore a better diagnostic tool than the use of serum markers separately. This is further confirmed by a positive likelihood ratio for the combination of CA125 and HE4 of 104.5 (95% CI 13.65–800.05) much higher than CA125 (6.97, 95% CI 3.91–12.41) and HE4 (5.81, 95% CI 2.83–11.90) alone. While predictive value is commonly used by clinicians, likelihood ratio is the probability that a person who is sick is a positive test divided by the probability that a non-sick person who also has the positive test. A positive likelihood ratio (LR+) > 10 means that the test is very good for evaluating the diagnosis [35]. This high LR+ values are comparable with some studies, including a recent study by Romagnolo et al [36]. Thus, combination of elevated CA125 and HE4 seems the useful diagnostic tool to confirm ovarian cancer in women with presumed benign ovarian tumour and can be used in addition to individual markers.

The RMI algorithm is a tool sometimes used in daily practice, given its simplicity. In our study, this algorithm has a high specificity of 99%, which is comparable to that described by Jacobs et al. (96.9%) [13]. Another study including nearly 1000 patients demonstrated that the assessment by ultrasound was superior according to IOTA criteria compared to the use of RMI [21]. The specificity of RMI for diagnosing ovarian cancer is rather high; found to be 92.4% in a meta-analysis [37] and 92.4% in a study by Van Gorp et al. [38]. We also find an AUC for RMI lower than the area of HE4, the ROMA algorithm or the CA125 and HE4 association (0.91, 0.92 and 0.92, respectively) which means that the RMI algorithm is not the best aid tool for the diagnosis of ovarian cancer.

For the ROMA algorithm, a significantly lower specificity (83.3%, 95% CI 77.5–88) was found compared to the other serum markers or to the RMI algorithm. In a meta-analysis, the ROMA algorithm was reported with less specificity than HE4 (84% vs. 94%), but with a better correlation than CA125 (84% vs. 78%) [23]. In another meta-analysis ROMA has a sensitivity between 76% and 86%, and a specificity between 74% and 95%, regardless of the method for assaying the markers [39]. The specificity of the ROMA algorithm (80.0%) and the AUC (0.92 (95% CI 0.87–0.98)) is comparable with some studies [32]. However, as its specificity is significantly lower than combination HE4 and CA125, ROMA is not the best tool to predict the risk of ovarian cancer.

Our results must be interpreted in light of certain limitations. First, the sample size calculation was done before the beginning of the study with an expected prevalence of ovarian cancers of 8.3%, compared to a preliminary study carried out in our service in a population finding five malignant lesions on 60 patients presenting a PBOT. In our study, we found only 12 malignant lesions among the 221 included women, leading to a prevalence of 5.4%. This prevalence is lower than our prediction, but more in accordance with the literature (4.9%) [40]. For this reason, likelihood ratio seems useful for outcomes assessment ratio. We therefore included only two women with adenocarcinoma, which may be weakly representative, but this is due to the very restrictive nature of our inclusion criteria, as there was no ascites, metastasis or tumour of malignant appearance on ultrasound. This allows us to be reassured about the role of ultrasound screening. Second, the sample size calculation was also done with an expected difference of specificity of 10% between HE4 and CA125. We failed to demonstrate a significant difference between the specificity of CA125 (90.4%) and that of HE4 (91.4%). Probably, the change in the inclusion criteria, since it only concerns patients with PBOT, has been able to modify the values found in the literature of the specificities of HE4 and CA125. On the other hand, we showed a significant difference in the specificity of the association of the two markers HE4 and CA125. In addition, this is the first study in the literature to evaluate CA125, HE4 and algorithms RMI and ROMA with strict inclusion criteria, such as those described by the IOTA group.

## 5. Conclusions

The combination of elevated HE4 and CA125 represents the best tool to confirm the risk of ovarian cancer in patients with a PBOT. With a specificity of 99.5% and a positive likelihood ratio of 104.5, the association of these two serum markers is better than serum markers used alone or than RMI or ROMA algorithms. Moreover, the AUC of 0.92 makes it an effective diagnostic test. To date, this is the first study with strict ultrasound inclusion criteria of PBOT as described by the IOTA group and thus definitively useful in the diagnostic procedure of ovarian cancer in women with ovarian tumour.

## Figures and Tables

**Figure 1 jcm-08-01784-f001:**
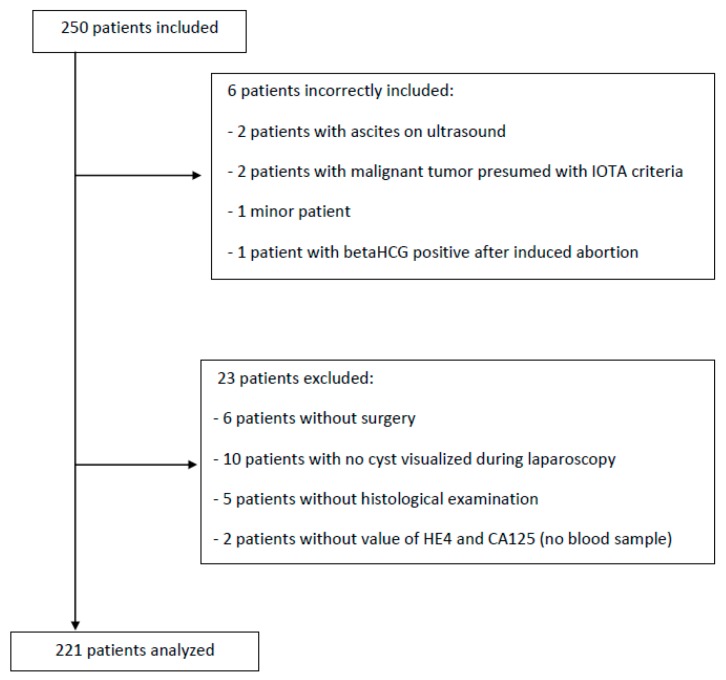
Flow chart.

**Figure 2 jcm-08-01784-f002:**
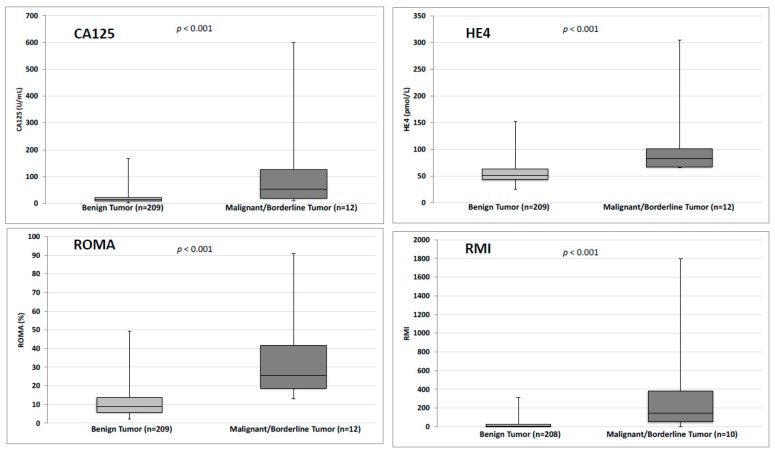
Values of HE4, CA125, RMI and ROMA algorithms in benign and malignant group. CA125: Carbohydrate Antigen 125. HE4: Human Epididymis Protein 4. RMI: Risk of Malignancy Index. ROMA: Risk of Ovarian Malignancy Index.

**Table 1 jcm-08-01784-t001:** Demographic data and serum markers and algorithm according to histological result.

	Benign Tumor (*n* = 209)	Borderline/Malign Tumor (*n* = 12)	Total *n* (%)	*p*
Age (years)	46.2 ± 15.1	53.1 ± 16.3	46.5 ± 15.2	0.14
(Minimum-Maximum)	(19–88)	(25–79)	(19–88)	
Body Mass Index (kg/m^2^)	24.6 ± 4.8	24.8 ± 5.7	24.6 ± 4.9	0.95
(Minimum-Maximum)	(14.9–39.8)	(17.3–35.5)	(14.9–39.8)	
CA125 < 35 U/mL	189 (90.4)	4 (33.3)	193 (87.3)	<0.001
HE4 < 70 pmol/L for premenopausal women OR < 140 pmol/L for postmenopausal women	191 (91.4)	6 (50)	197 (89.1)	<0.001
RMI score > 200	2 (1)	4 (40)	6 (2.8)	<0.001
ROMA score > 11.4% for premenopausal women OR > 29.9% for postmenopausal women	35 (16.7)	9 (75)	44 (19.9)	<0.001

Data are mean ± standard deviation or *n* (%) unless otherwise specified. Student’s *t* test, χ^2^ test, nonparametric Mann–Whitney test, and Fisher’s exact test were used as appropriate. A *p*-value less than 0.05 was considered significant.

**Table 2 jcm-08-01784-t002:** Comparisons of the specificity of serum markers and algorithms.

	CA125 (Sp = 90.4%)	HE4 (Sp = 91.4%)	CA125 + HE4 (Sp = 99.5%)	RMI (Sp = 99.0%)	ROMA (Sp = 83.3%)
**CA125 (Sp = 90.4%)**					
**HE4 (Sp = 91.4%)**	0.87				
**CA125 + HE4 (Sp = 99.5%)**	< 0.001	< 0.001			
**RMI (Sp = 99.0%)**	< 0.001	< 0.001	1		
**ROMA (Sp = 83.3%)**	0.04	< 0.001	< 0.001	< 0.001	

Data are *p* value. A *p*-value less than 0.05 was considered significant. Sp: specificity.

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
