# Peer review of "Efficacy of HE4, CA125, Risk of Malignancy Index and Risk of Ovarian Malignancy Index to Detect Ovarian Cancer in Women with Presumed Benign Ovarian Tumours: A Prospective, Multicentre Trial"

_jcm, 2019, doi:10.3390/jcm8111784_

Round 1

Reviewer 1 Report

In this paper Vincent Dochez et al. evaluated the Efficacy of HE4, CA125, RMI and ROMA to detect ovarian cancer in women with Presumed Benign Ovarian Tumours.

Although this manuscript is well-organized and comprehensively described, however it is too simplistic and lacks technical substance. The Authors employed the simple statistical test and limited numbers of local patients to confirm the proven results. Actually, there is no novelty in the approach (neither experimental design nor statistical method) and therer is redundancy with other results, e.g., "Serum human epididymis protein 4 vs carbohydrate antigen 125 for ovarian cancer diagnosis: a systematic review" and "Does risk for ovarian malignancy algorithm excel human epididymis protein 4 and CA125 in predicting epithelial ovarian cancer: a meta-analysis".

Author Response

Reviewer 1:

In this paper Vincent Dochez et al. evaluated the Efficacy of HE4, CA125, RMI and ROMA to detect ovarian cancer in women with Presumed Benign Ovarian Tumours.

Although this manuscript is well-organized and comprehensively described, however it is too simplistic and lacks technical substance. The Authors employed the simple statistical test and limited numbers of local patients to confirm the proven results. Actually, there is no novelty in the approach (neither experimental design nor statistical method) and therer is redundancy with other results, e.g., "Serum human epididymis protein 4 vs carbohydrate antigen 125 for ovarian cancer diagnosis: a systematic review" and "Does risk for ovarian malignancy algorithm excel human epididymis protein 4 and CA125 in predicting epithelial ovarian cancer: a meta-analysis".

Thank you for your synthesis and your comments. Unlike the articles cited in the literature, our study is based on patients with PBOT. This is the only and first study currently in the literature focusing on this very interesting population, because it does not count tumours suggestive of cancer and therefore requiring surgical management. Moreover, our article describes the association HE4 and CA125 that few articles quote in the literature. So we think our manuscript is relevant to your journal.

Reviewer 2 Report

In my opinion, the present study is well-conceived and can be accepted for publication. The results are important for better assessment of risk for development of ovarian cancer. I have only few remarks listed below, and I would also recommend to improve English (or one may say presentation/writing) in the manuscript.

Remarks:
- L64: The authors explained well CA125 and HE4, but they haven't explained anyhow RMI and ROMA. No references were also provided. But that line already states the use of RMI and ROMA. I think the explanation for latter two should be given before L64 and be a part of introduction. Otherwise it is unclear.
- L71: Could the authors provide the reasoning why women age < 18 were excluded?

Typos, etc.
- Abstract: (inconsistency) I didn't understand why the acronym is ROMA and not ROMI. Second remark, why "two hundred and fifty patients", but then "221" is written with numbers? Third, it should be "Conclusion*s*" not "ConclusionS".

Author Response

Reviewer 2:
In my opinion, the present study is well-conceived and can be accepted for publication. The results are important for better assessment of risk for development of ovarian cancer.

Thank you for your comments and your relevant summary.

I have only few remarks listed below, and I would also recommend to improve English (or one may say presentation/writing) in the manuscript.

Remarks:
- L64: The authors explained well CA125 and HE4, but they haven't explained anyhow RMI and ROMA. No references were also provided. But that line already states the use of RMI and ROMA. I think the explanation for latter two should be given before L64 and be a part of introduction. Otherwise it is unclear.

We had reviewed the paper by a native-US English biomedical editor. The references had been quoted later in the text. We have advanced them, as the definition of these algorithms.

- L71: Could the authors provide the reasoning why women age < 18 were excluded?

It is always difficult in a prospective study to obtain the consent of minor patients, as it also requires parental consent. In addition, PBOT are quite rare in these minor patients, and therefore this allows for better external validity.

Typos, etc.
- Abstract: (inconsistency) I didn't understand why the acronym is ROMA and not ROMI. Second remark, why "two hundred and fifty patients", but then "221" is written with numbers? Third, it should be "Conclusion*s*" not "ConclusionS".

We have corrected the sentences. The acronym RMI was introduced by Jacobs et al. in 1990. It is therefore not possible for us today to rename it from RMI to ROMI. Similarly ROMA has been described by Moore et al. in 2009.

Reviewer 3 Report

The first line in the in abstract is not clear. please rephrase this sentence In the line 19 page1 , the author should mention the full form of IOTA. The author should be consistent with the subheading inside the Abstract. For example, Methods: is in Caps but other are not. In line 47, page 2, author should mention some serum marker which has been evaluated. In introduction author should give brief description about RMI and ROMA algorithms. Page 3, line 111, author should describe the ECLIA method in this section.

Author Response

Reviewer 3:
The first line in the in abstract is not clear. please rephrase this sentence

The sentence has been corrected. We hope that these changes improve comprehension.

In the line 19 page1 , the author should mention the full form of IOTA.

The sentence has been corrected.

The author should be consistent with the subheading inside the Abstract. For example, Methods: is in Caps but other are not.

The format of subheading inside the abstract has been changed.

In line 47, page 2, author should mention some serum marker which has been evaluated.

The sentence has been corrected.

In introduction author should give brief description about RMI and ROMA algorithms.

We have specified RMI and ROMA algorithms in introduction lines 62-65.

Page 3, line 111, author should describe the ECLIA method in this section.

The ECLIA method has been described Page 3, line 105.

Reviewer 4 Report

The authors performed a prospective, observational, multicentre, laboratory-based study concerning biomarkers and ovarian tumors, and found that the malignant group had significantly higher mean values of HE4, CA125, RMI and ROMA compared to the benign group. Specificity was significantly higher using a combination of HE4 and CA125 compared to either HE4 or CA125 alone.

Could it be possible to give some background of RMI and ROMA in the Introduction? It will be easier for reader to understand why authors want to evaluate the efficacy to predict ovarian cancer in women presenting PBOT.

Author Response

Reviewer 4:
The authors performed a prospective, observational, multicentre, laboratory-based study concerning biomarkers and ovarian tumors, and found that the malignant group had significantly higher mean values of HE4, CA125, RMI and ROMA compared to the benign group. Specificity was significantly higher using a combination of HE4 and CA125 compared to either HE4 or CA125 alone.

Could it be possible to give some background of RMI and ROMA in the Introduction? It will be easier for reader to understand why authors want to evaluate the efficacy to predict ovarian cancer in women presenting PBOT.

Thank you for your comments and your relevant summary. We have specified RMI and ROMA algorithms in introduction lines 62-65.

Round 2

Reviewer 1 Report

Manuscript is well-improved and I hope it's worth to be published.